# Assessing the Asynchrony Event Based on the Ventilation Mode for Mechanically Ventilated Patients in ICU

**DOI:** 10.3390/bioengineering8120222

**Published:** 2021-12-18

**Authors:** Nur Sa’adah Muhamad Sauki, Nor Salwa Damanhuri, Nor Azlan Othman, Belinda Chong Chiew Meng, Yeong Shiong Chiew, Mohd Basri Mat Nor

**Affiliations:** 1School of Electrical Engineering, College of Engineering, Universiti Teknologi MARA, Cawangan Pulau Pinang, Permatang Pauh 13500, Malaysia; saadah1537@uitm.edu.my (N.S.M.S.); azlan253@uitm.edu.my (N.A.O.); belinda.chong@uitm.edu.my (B.C.C.M.); 2School of Engineering, Monash University Malaysia, Bandar Sunway 47500, Malaysia; Chiew.Yeong.Shiong@monash.edu; 3Department of Anaesthesiology and Intensive Care, School of Medicine, International Islamic University of Malaysia, Kuantan 25200, Malaysia; m.basri@iium.edu.my

**Keywords:** mechanical ventilation, lung elastance, asynchrony events, spontaneously breathing, ventilation mode

## Abstract

Respiratory system modelling can assist clinicians in making clinical decisions during mechanical ventilation (MV) management in intensive care. However, there are some cases where the MV patients produce asynchronous breathing (asynchrony events) due to the spontaneous breathing (SB) effort even though they are fully sedated. Currently, most of the developed models are only suitable for fully sedated patients, which means they cannot be implemented for patients who produce asynchrony in their breathing. This leads to an incorrect measurement of the actual underlying mechanics in these patients. As a result, there is a need to develop a model that can detect asynchrony in real-time and at the bedside throughout the ventilated days. This paper demonstrates the asynchronous event detection of MV patients in the ICU of a hospital by applying a developed extended time-varying elastance model. Data from 10 mechanically ventilated respiratory failure patients admitted at the International Islamic University Malaysia (IIUM) Hospital were collected. The results showed that the model-based technique precisely detected asynchrony events (AEs) throughout the ventilation days. The patients showed an increase in AEs during the ventilation period within the same ventilation mode. SIMV mode produced much higher asynchrony compared to SPONT mode (*p* < 0.05). The link between AEs and the lung elastance ($AUC\;E_{drs})\;$was also investigated. It was found that when the AEs increased, the $AUC\;E_{drs}\;$decreased and vice versa based on the results obtained in this research. The information of AEs and $AUC\;E_{drs}\;$provides the true underlying lung mechanics of the MV patients. Hence, this model-based method is capable of detecting the AEs in fully sedated MV patients and providing information that can potentially guide clinicians in selecting the optimal ventilation mode of MV, allowing for precise monitoring of respiratory mechanics in MV patients.

## 1. Introduction

A mechanical ventilation asynchrony event (AE) is also known as patient–ventilator asynchrony (PVA). It occurs when a patient’s respiratory effort does not match the mechanical ventilator’s breathing assistance [1,2]. AEs can happen at any time during full or partial mechanical ventilation (MV) [3]. Unfortunately, if more frequent AEs occur during MV therapy, it may result in poor patient–ventilator interaction or worsen the patient’s condition [4]. Therefore, it is important to have a mathematical model that can estimate the occurrences of AEs for both fully and partially mechanically ventilated patients suitable for real-time bedside application. Thus, the focus is to develop a model-based method that can detect asynchrony automatically for a clinical dataset. In addition, the secondary objective is to apply the model and study the AEs in ICU cohorts in Southeast Asia.

There are some model-based methods that have been developed to estimate the AEs in MV patients. For instance, work by Sinderby et.al, who have studied on the patient–ventilator interaction by comparing between the ventilator pressure and diaphragm electrical activity (EAdi) waveforms [5]. The authors claimed that the created approach is more sensitive to analysis and provides greater precision; nevertheless, the method necessitates the use of extra instruments to measure and gather EAdi data, which adds to the expense. Hence, it is essential to have a non-invasive mathematical model that can be applied to analyze and detect PVA in fully sedated and partially ventilated patients, at the bedside and without adding invasive measurements or protocols.

A number of studies have demonstrated the ability to detect AEs in MV patients without using additional invasive equipment [6,7]. Several research in recent years have created a computational approach to eliminate the need for extra equipment or protocols to detect AEs in MV patients. For example, study by Blanch et al. [8]. According to the authors, there is a prevalence of different asynchronies in MV patients that occur throughout MV therapy, and that these asynchronies might affect patients’ outcomes. Due to that, the authors had conducted a study in order to assess the prevalence of five types of asynchronies: ineffective inspiratory efforts during expiration (IEE), double-triggering (DT), aborted inspirations, short cycling, and prolonged cycling in MV patients. The authors declared that the most common asynchrony occurs in all ventilation modes (PCV, VCV, and PSV) was IEE. However, there is contrast findings to the study by Zhang et.al [9]. The authors have proposed a computerized algorithm, 2-layer long short-term memory (LSTM) network, to identify the AEs in MV patients. The authors revealed that the DT, rather than the IEE, is the most commonly encountered, with scores of 0.983 and 0.979. Yet, the tool of detection AEs in MV patients is still limited. As the patient breathing has inconsistency and there is still a need for the clinicians to further check on the patient–ventilator asynchrony.

Currently, the standard for analyzing patient–ventilator interaction is through calculating the asynchrony index (AI) [10,11]. AI is described as a percentage of the estimated total number of AEs over the total number of breathing cycles [6,7]. Recently, Loo et al. had developed a machine learning method to detect the presence of AEs [12]. The authors noted that a convolutional autoencoder model was able to recognize and quantify the presence of AEs. The model was trained to familiarize itself with the normal breathing cycle of MV patients. The model then determines the severity of AEs by comparing the identified normal breathing waveform to the altered waveform. Even though the model showed ability to detect and quantify AEs, the model had only been tested on AEs observed in the airway pressure waveform. As asynchrony can occur in any MV modes, asynchrony detection should be incorporated with the analysis of both airway pressure and flow. However, the proposed method only quantifies the AEs in SIMV mode, as the patients have been ventilated using MV in various ventilation mode [13].

Furthermore, asynchronous breathing also increases the work of breathing, prolongs the length of MV and increases the MV [6,8,14]. Thus, it is essential to have a non-invasive model that can monitor the asynchrony event in MV patients that will help estimate the respiratory mechanics of the patients.

In this study, a non-invasive model-based method was developed to assess and evaluate the asynchrony event in mechanically ventilated patients based on the data collected from the International Islamic University (IIUM) Hospital, Malaysia. This method was developed based on time-varying elastance model [15], and in this paper, the relationship between asynchrony and the ventilation mode was further investigated [15,16]. This method allows for the estimation of asynchrony events and potentially provides unique insight regarding the severity of asynchronous breathing that can be used to manage MV better.

## 2. Methodology

### 2.1. Patient Data

Data from 10 patients were admitted to the Intensive Care Unit (ICU) at the International Islamic University Malaysia (IIUM) Hospital were used in this work. The data were based on the inclusion and exclusion criteria mentioned in the Clinical Application of Respiratory Elastance (CARE trial) software system [17]. The patients were ventilated using Puritan Bennet PB980 ventilators under a ventilation mode as determined by the clinicians. Two ventilation modes were employed in this study: (1) synchronized intermittent mandatory ventilation (SIMV) with volume-controlled/pressure-controlled ventilation (VCV/PCV) mode and (2) spontaneous breathing (SPONT) with proportional assist ventilation (PAV) mode.

The IIUM Research Ethics Committee (IREC) had approved the trial and the use of the patients’ data. The ethics number was IREC666 and the trial was also registered with Clinical Trial Registry New Zealand of Australia (ANZCTR). Table 1 shows the patient demographics.

### 2.2. Time-Varying Elastance Model

Patient-specific respiratory elastance of the MV patients, was estimated using the time varying elastance $(E_{drs})\;$model that was developed by Chiew et al. [15]. Time-varying elastance model is defined as follows:(1)$$P_{aw}\left(t\right)=R_{rs}Q_{aw}\left(t\right)+E_{drs}\left(t\right)V\left(t\right)+P_{0}\;$$
where $P_{aw}$, is the airway pressure, *t* is time, $E_{drs}$ is time-varying elastance, *V* is volume, $R_{rs}$ is airway resistance, $Q_{aw}$ is flow and $P_{o}$ is the offset pressure or PEEP. An integral based-method is used to estimate the airway resistance $R_{rs}$ and elastance $E_{drs}$ of MV patient [18,19], as follows:(2)$$\int^{\;}P_{aw}\left(t\right)dt=R_{rs}\int^{\;}Q_{aw}\left(t\right)dt+E_{drs}\left(t\right)\int^{\;}V\left(t\right)dt+\int^{\;}P_{o}\left(t\right)dt$$

The respiratory elastance $E_{drs}$ of MV patient can be estimated using a constant value of lung resistance $R_{rs}$ of patient specific. The calculated $E_{drs}\;$best fit is described as:(3)$$E_{drs}\left(t\right)=\frac{P_{aw}\left(t\right)-P_{o}-\left(R_{rs}Q_{aw}\left(t\right)\right)}{V\left(t\right)}$$

Then, from Equation (3), the area under the curve of $E_{drs}$ $\left(AUC\;E_{drs}\right)$, as seen in Equation (4), is calculated to allow the comparison for each breathing cycle [20]:(4)$$AUC\;E_{drs}=\frac{\int^{\;}E_{drs}\left(t\right)dt}{t}\;$$

### 2.3. Asynchrony Detection

As mentioned by Poole. et al., the detection presence of asynchrony event (AE)s of MV patients was thru + 50% above the median ($AUC\;E_{drs}$) over each breathing cycle of a given patient [16]. The detection of AEs is described as follows:(5)$$Detection\;of\;AEs=+\frac{150}{100}\times Median\;of\;AUC\;E_{drs}\;$$

The asynchrony index (AI) for each patient is calculated using the following equation:(6)$$Asynchrony\;Index=\frac{Total\;AEs}{Total\;Breathing\;Cycle}\times 100\%$$

### 2.4. Data Analysis

In this study, the $E_{drs\;}$for each patient’s breathing cycle with various ventilation modes were calculated. The data were presented as the median and interquartile range (IQR) of ($AUC\;E_{drs}$) to identify the asynchrony events of the MV patients throughout the ventilated days as well as the type of ventilation modes.

## 3. Results

Table 2 illustrates the analyzed data for 10 MV patients for this study. The summary for the analysis of the detection of AEs and AI in each ventilation mode is tabulated in Table 2. They are also plotted in the boxplot in Figure 1 to compare the number of AEs and AI in each ventilation mode.

Figure 1 depicts two boxplots for the total number of AEs and AI for all patients that occur at different ventilation modes. It can be seen that there is a significant difference in the number of AEs occur during the MV therapy at different ventilation modes. The patients who are ventilated in SIMV mode show a higher number of asynchrony events compared to the SPONT mode, (*p* < 0.05). However, there is not a significant difference in AI observed in each ventilation mode, (*p* > 0.05).

Figure 2 and Figure 3 depict the analysis of $P_{aw}$, $Q_{aw}\;$and $AUC\;E_{drs}\;$for Patients 2 and 8, respectively. As plotted in Figure 2, a flat and brief $AUC\;E_{drs}$, is observed when there is no inconsistent shape in the analyzed airway pressure and flow. As a result, there are no asynchrony events detected. Based on Figure 3, it is observed that there is an ill-matched shape shown in the analysis of airway pressure and flow. It can be clearly seen that there are a few breathing cycles above the 150% median $AUC\;E_{drs}$ threshold, suggesting that AEs occurs due to the spontaneous breathing efforts produced by the patient.

Figure 4 shows the box plot of $AUC\;E_{drs}$ plotted according to ventilation days for Patients 1, 2, 4, 5, 7 and 9, respectively. The Cumulative Distribution Functions (CDFs) of Patients 3 and 10 are shown in Figure 5, along with dashed lines indicating the 95% confidence intervals (5th and 95th percentile) of $AUC\;E_{drs}\;$in each patient. Figure 6 and Figure 7 show the corresponding boxplot for Patients 6, 8 and 9 of $AUC\;E_{drs}$ versus the ventilation modes. Each patient exhibits different $AUC\;E_{drs}$ under different ventilation days and ventilation modes.

## 4. Discussion

The respiratory elastance,$\;E_{rs},\;$is one key metric that can be used to detect and quantify the asynchrony events (AEs) of MV patients [21,22]. AEs appear when the patient’s breathing effort is not synchronized with the mechanical ventilation’s breathing support, resulting in a mismatch between airway pressure and flow. Hence, the AEs affects the estimation of the true underlying respiratory mechanics in MV patients. Thus, the estimation of AEs is essential for better management of MV.

Table 2 tabulates the summarized results from all 10 MV patients in terms of detection of AEs based on estimation of $AUC\;E_{drs}\;$and mechanical ventilation mode according to days. It can be clearly seen from Table 2 that all patients have demonstrated AEs throughout the ventilated days, with Patient 6 having the highest AEs of 509 and AI of 40.36% during SIMV mode on Day 2 of ventilation. From the summarized data, it can be concluded that AEs can arise at any moment during the ventilation period. According to the findings obtained in this study, patients who are ventilated in SIMV mode have a higher number of AEs and AI than those ventilated in the SPONT mode, as depicted in Figure 1.

As illustrated in Table 2, it is found that Patients 1, 2, 4, 5, 6, 7, 8 and 9 show increased numbers of AEs from Day 1 to Day 2. As depicted in Figure 4, it is also found that based on the $AUC\;E_{drs}\;$of the same patients, Patients 1, 2, 4, 5, 6, 7, 8 and 9 have decreased the value of $AUC\;E_{drs}$ on Day 2 of ventilation. These results indicate that there is some improvement in recruitment of the lung since the value of $AUC\;E_{drs}$ has decreased [23,24]. This increased number of AEs and decreased of $AUC\;E_{drs}$ could also be due to the breathing efforts made by the patients. These findings suggest that an increase in the number of AEs and a decrease in $AUC\;E_{drs}\;$may be associated with the patients regaining their spontaneous breathing effort and beginning to “fight” against ventilator assistance.

Dissimilar findings were obtained from Patients 3 and 10, who were ventilated in the SIMV mode during the MV therapy. On the first day of ventilation, these patients have shown some improvement during the MV therapy based on the decreasing value of $AUC\;E_{drs}$ as a lower elastance value indicates that there is some improvement and recruitment of the lung as shown in Table 2. In contrast, the number of AEs are higher for both Patients 3 and 10 on the first day of ventilation. However, the estimation value of $AUC\;E_{drs}$ has increased for both patients with a lower detection of AEs on the second day of ventilation, as shown in Table 2.

Furthermore, from the CDF plot in Figure 5, Patient 3 has a higher range of $AUC\;E_{drs}$ with 22.58 [19.37–26.73] cmH_2_O·s/L on day 1 of ventilation. For example, the 95% confidence interval (5th and 95th percentile) value is significantly larger on Day 1 compared to Day 2 of ventilation, although both days have an almost similar median value of 22.58 cmH_2_O·s/L and 22.79 cmH_2_O·s/L, respectively. Similarly, for Patient 10, the CDF plot shown in Figure 5 depicts that the 95% confidence interval of $AUC\;E_{drs}$ for this patient is significantly larger in Day 1 compared to Day 2 of ventilation. These results are as expected as both patients exhibit a significantly higher asynchrony on Day 1 compared to Day 2 of ventilation. Even though the form of the distribution $AUC\;E_{drs}$ varies from day to day, but the results show that the tendency to detect the number of AEs is the same. This is because it shows that this model is capable of capturing the differences in respiratory mechanics and response to MV. However, based on the results, there is no observable trend between $AUC\;E_{drs}\;$and AEs. The increase in $AUC\;E_{drs}\;$can relate to the patient’s condition as a lower effective elastance implies less risk of lung damage [15,25,26,27]. It can be concluded that a higher value of $AUC\;E_{drs}\;$Indicates the overdistension of the lung condition of the patient.

As illustrated in Figure 5, there are occurrences of negative values due to negative elastance, $E_{drs}\;$produced by Patient 10. Negative elastance has been reported to be one of the measures that can be used to detect asynchrony effort made by patients during MV therapy [28]. Thus, there is a need for further investigation into the distribution of negative elastance in MV patients.

Based on the results depicted in Table 2, Patients 6, 8 and 9 have been ventilated in two types of ventilation modes: SIMV and SPONT for 3 days of ventilation. According to the results tabulated in Table 2, Patients 6, 8 and 9 show a lower number of AEs and AI detected by the model-based method on the 1st day of ventilation. On the 2nd day of ventilation, the number of AEs and AI increases and the value of $AUC\;E_{drs}$ decreases, as illustrated in Figure 6 and Figure 7. However, on the third day of ventilation, only Patients 6 and 8 show decreased numbers of AEs and increased values for $AUC\;E_{drs},\;$respectively. The increase in $AUC\;E_{drs}$ and the decrease number of AEs could be attributed to clinicians changing the ventilation mode during the MV therapy.

Table 2, however, demonstrates the contrast finding for Patient 9 in comparison to Patients 6 and 8. Patient 9 appears to have a further decrease in the value of $AUC\;E_{drs},\;$despite the fact that the detection of AEs decreases as the ventilation mode changes from SIMV mode to SPONT mode. A consistent decreasing trend would indicate that the patient’s lung condition is improving, as shown in Figure 7. With these findings, this study provides evidence that adjusting the ventilation mode could either reduce or increase the number of AEs and value of $AUC\;E_{drs\;}\;$throughout the ventilated days. The finding is consistent with the findings of a past study conducted by Poole et al. suggesting that changing the ventilator mode might either improve or aggravate the patient’s condition [16].

### 4.1. The Importance of Real-Time Assessment in AEs and $AUC\;E_{drs}$

According to the results illustrated in Table 2, all patients show an increase in AEs and a decrease value of $AUC\;E_{drs}\;$during the ventilation period using the same ventilation mode. Hence, these findings highlight the necessity for attending clinicians to change the ventilation mode based on the AEs during the ventilation. This could potentially improve patient–ventilator interaction and care management.

The transition from SIMV to SPONT in Patients 6 and 8 shown in Figure 6 resulting in a decreased number of AEs and increment value of $AUC\;E_{drs}$. The increase of $AUC\;E_{drs}$ obtained in this study suggests that there is a need to maintain the same ventilation mode. This is on the condition that the value of $AUC\;E_{drs}\;$decreases day by day as depicted in Figure 7 or Patient 9. This suggests that the lung condition of this patient is showing signs of improvement, and the patient can be switched to another ventilation mode that is more suitable to the patient’s condition since increasing of AEs indicates that the patient may have the ability to breathe spontaneously [22].

The primary issue that is raised in this study is the lack of a feasible method for identifying these asynchrony levels using both analysis of airway pressure and flow in real-time. As a result, clinicians are unable to analyze this element of the patient–ventilator interaction. Patients are also at risk of prolonged ventilation and other unfavorable consequences due to a lack of clinical diagnostic tools. Hence, based on the result as presented here, it has demonstrated that the $AUC\;E_{drs}\;$metric is capable of detecting AEs in ventilated patients. This non-invasive method is able to provide specific information about the patient’s progress, as well as guide and help clinicians to optimize the setting of the MV.

### 4.2. Limitations

Although the suggested model-based detection of AEs in this study is capable of capturing asynchrony levels throughout the ventilation time, its predictive potential is limited. This proof-of-concept study is thus a first step in investigating the feasibility of this model-based detection of the AEs technique. However, the study may not have the capacity to say AE is more prevalent in which ventilation mode. There is no significant difference found, and even with the different trends observed in AE in different ventilation modes, more research is required to understand the fundamental physiologic response in our patients. Furthermore, in this patient data set, there is no patient breathing triggers detected during MV. To further prove the accuracy of patient–ventilator asynchrony detection for MV patients, more patient data are needed to demonstrate the complete robustness of the AE detection method. However, the currently limited patient data set were appropriate for testing the strategy and demonstrating its potential value in detecting asynchrony levels in MV patients.

## 5. Conclusions

Patients who are experiencing shortness of breath require MV for breathing support. During the MV therapy, the patients improve, especially those who have altered the airway pressure and airflow flow due to their breathing efforts. Hence, this situation will lead to an erroneous estimation of the true and accurate respiratory mechanics, which will eventually lead to further lung damage. Thus, this paper investigates the asynchronous event of MV patients in ICU hospitals by adopting a developed extended time-varying elastance model. The model-based technique can be used to consistently detect and quantify the AEs for MV patients who are ventilated in various ventilation modes. The relation of AEs and $AUC\;E_{drs}\;$has also been investigated where there is no observable trend between $AUC\;E_{drs}\;$and AEs. This real-time metric is straightforward to implement and is computationally simple to implement in real-time at the bedside without requiring significant clinical processes. Clinically, this approach may influence MV steering and provide unique information that clinicians can use to select the proper ventilator settings and enhance the patient’s condition.

## Figures and Tables

**Figure 1 bioengineering-08-00222-f001:**
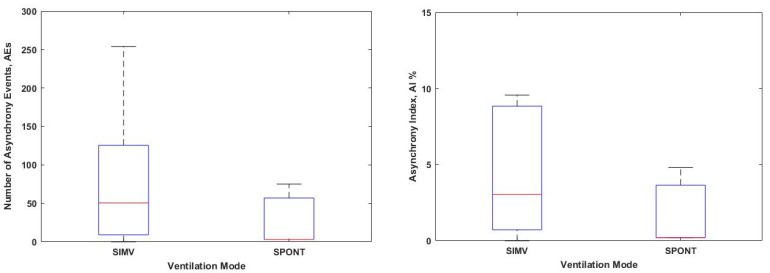
The total number of (**Left**) asynchrony events and (**Right**) the asynchrony index for each ventilation mode for all patients are plotted in a boxplot.

**Figure 2 bioengineering-08-00222-f002:**
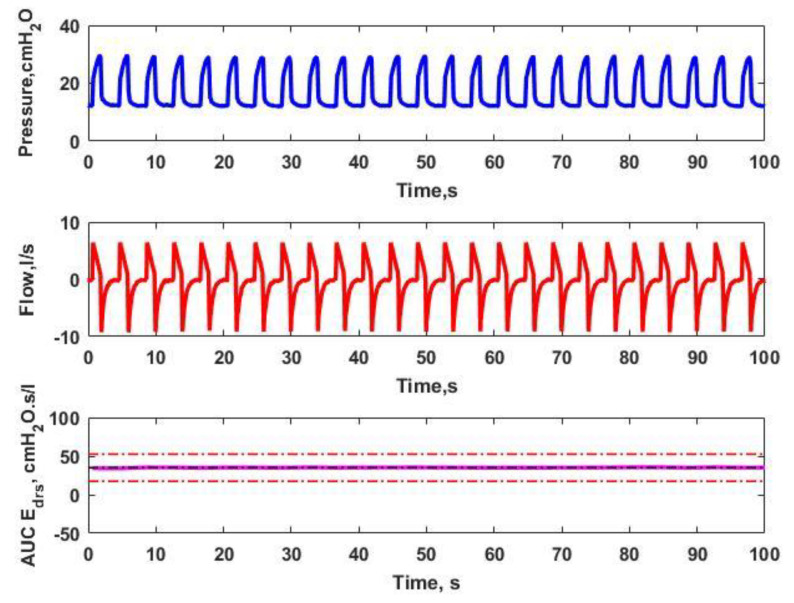
Subdivisions of $P_{aw}$, $Q_{aw}\;$ and $AUC\;E_{drs}\;$ for Patient 8. There are no inconsistent shapes shown in the airway pressure and airway flow. Thus, this results in a smooth and transient $AUC\;E_{drs}$, in which there is no AEs occurring for Patient 8.

**Figure 3 bioengineering-08-00222-f003:**
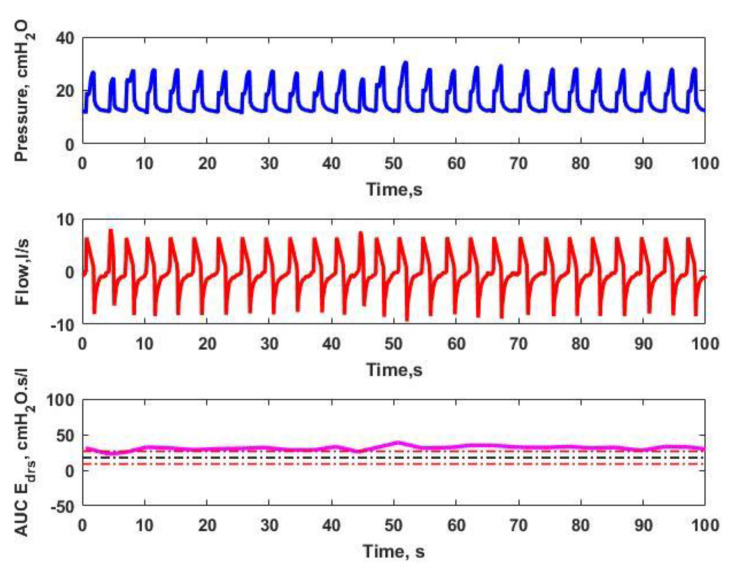
Subdivisions of $P_{aw}$, $Q_{aw}\;$ and $AUC\;E_{drs}\;$ for Patient 8, containing AEs which resulting in a sudden change of $AUC\;E_{drs}\;$ which indicates that AEs have occurred.

**Figure 4 bioengineering-08-00222-f004:**
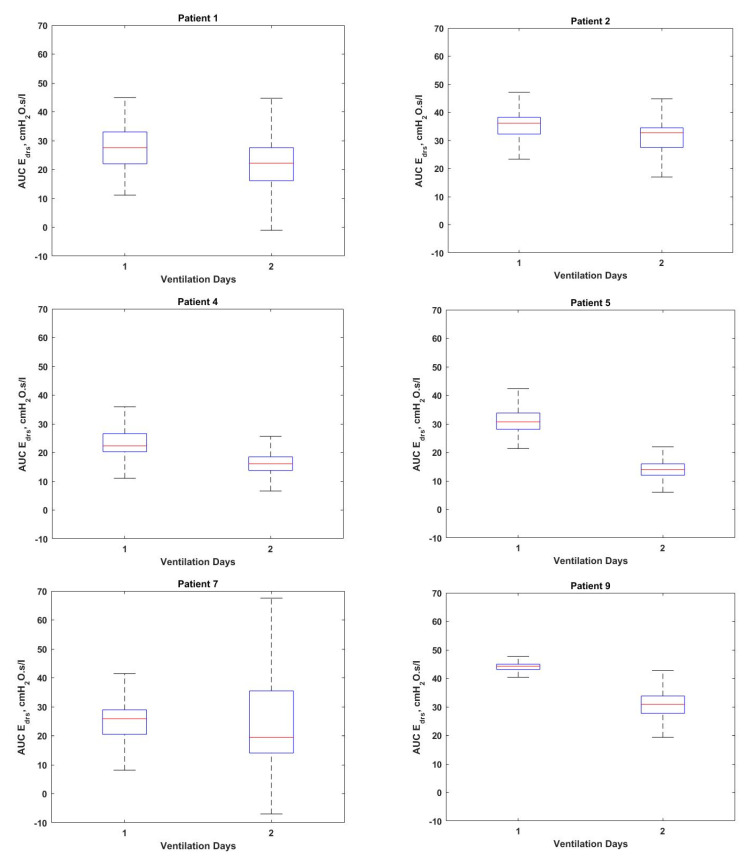
The distribution of $AUC\;E_{drs}$ by ventilation days for (**Left-Top**) Patient 1 (**Right-Top**) Patient 2 (**Left-Middle**) Patient 4 (**Right-Middle**) Patient 5 (**Left-Bottom**) Patient 7 and (**Right-Bottom**) Patient 9.

**Figure 5 bioengineering-08-00222-f005:**
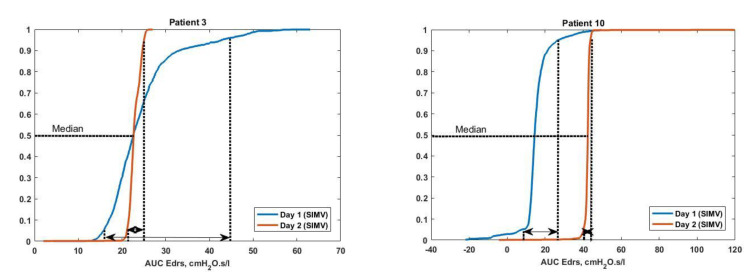
$AUC\;E_{drs}\;$cumulative distribution function (CDF) plotted by ventilation days in SIMV ventilation mode for (**Left**) Patient 3 (**Right**) Patient 10. The dashed lines show the 95% confidence interval (5th and 95th percentile) of $AUC\;E_{drs}$.

**Figure 6 bioengineering-08-00222-f006:**
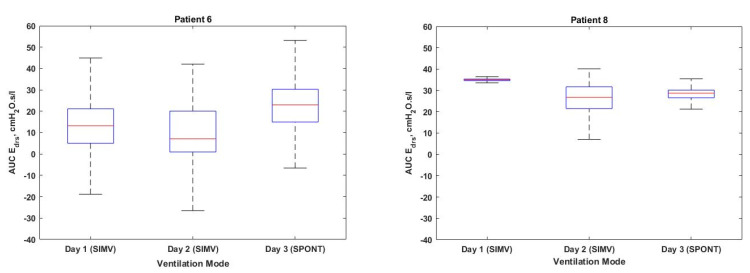
The classifications of $AUC\;E_{drs}\;$by ventilation mode for (**Left)** Patient 6 and (**Right**) Patient 8.

**Figure 7 bioengineering-08-00222-f007:**
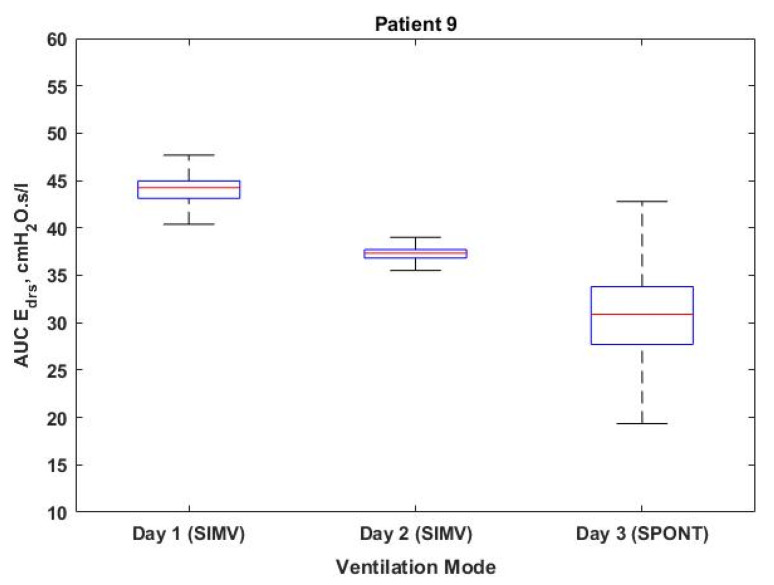
Boxplot of $AUC\;E_{drs}\;$for Patient 9 in different ventilation modes.

**Table 1 bioengineering-08-00222-t001:** Characteristics of the patients.

Patient No	Gender	Age	Clinical Diagnosis
1	Female	64	Pneumonia
2	Female	34	Pneumonia
3	Male	43	Pneumonia
4	Male	74	Pneumonia
5	Male	48	ARDS
6	Female	43	Thyroid
7	Male	52	CA Lung and SVC Obstruction
8	Male	64	Respiratory Failure, HAP, ESRF
9	Female	66	Septic shock 2° to HAP with Bronchospasms
10	Female	63	Septic shock

**Table 2 bioengineering-08-00222-t002:** Asynchrony events and asynchrony index by days for each patient based on ventilation mode and $AUC\;E_{drs}$ across all PEEP levels.

Patient No	Day	Ventilation Mode	Breathing Cycle	No of AEs	AI %	AUC $E_{drs}$ Median [IQR] (cmH_2_O·s/L)	PEEP (cmH_2_O)
1	1	SIMV VCV	1370	14	1.02	27.59 [21.98–33.00]	3–5
	2	SIMV VCV	1853	254	13.71	21.97 [15.36–27.78]	12–19
2	1	SIMV PCV	1469	32	2.18	36.15 [32.27–38.22]	8–9
	2	SIMV PCV	1816	43	2.37	32.75 [27.50–34.52]	15–17
3	1	SIMV VCV	1321	124	9.39	22.58 [19.37–26.73]	9–10
	2	SIMV VCV	1380	0	0	22.79 [22.11–24.26]	10–11
4	1	SIMV VCV	1461	94	6.43	22.02 [20.11–26.12]	8–18
	2	SIMV VCV	1349	129	9.56	16.16 [13.82–1860]	6–15
5	1	SIMV VCV	1473	6	0.41	30.67 [27.86–34.02]	11–12
	2	SIMV VCV	1389	115	8.28	13.79 [11.28–15.83]	10–12
6	1	SIMV VCV	1418	452	31.88	12.10 [3.90–20.25]	12–14
	2	SIMV VCV	1261	509	40.36	7.11 [0.92–20.09]	12–13
	3	SPONT PAV	1564	75	4.80	23.32 [15.72–30.28]	10–12
7	1	SIMV PCV	1258	58	4.61	25.88 [20.37–2897]	8–17
	2	SIMV PCV	1077	405	37.60	19.47 [14.08–35.48]	10–14
8	1	SIMV VCV	1240	0	0	35.01 [34.56–35.44]	12–13
	2	SIMV VCV	1258	20	1.59	24.89 [19.22–30.90]	12–13
	3	SPONT PAV	1602	3	0.19	28.57 [26.33–29.81]	8–10
9	1	SIMV VCV	1188	0	0	44.16 [43.10–44.92]	12–13
	2	SIMV VCV	1160	12	1.03	37.34 [36.82–37.71]	12–13
	3	SPONT PAV	1456	3	0.21	32.04 [28.59–35.19]	12–13
10	1	SIMV PCV	1645	127	7.72	14.26 [12.88–16.25]	16–31
	2	SIMV PCV	1314	2	0.15	42.45 [41.91–42.86]	10–11

## Data Availability

Data can be obtained upon request by contacting the corresponding author at norsalwa071@uitm.edu.my.

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
