# Peer review of "Assessing the Asynchrony Event Based on the Ventilation Mode for Mechanically Ventilated Patients in ICU"

_bioengineering, 2021, doi:10.3390/bioengineering8120222_

Round 1
Reviewer 1 Report
Sauki et al propose a method to identify an asynchronic breathing pattern in MV patients in ICU. The paper is very difficult to follow because is not a complete math paper and the clinical translation of the findings is incomplete and difficult to understand in many parts of the manuscript. Moreover, important precedent investigations on the field are lacking. My comments:
Introduction first paragraph. The description of what an asynchrony is differs substantially of general understanding. For instance, asynchronies might or might not be related with the intensity of respiratory effort. Please revise.
Intro second paragraph: “There are some model-based methods that have been developed to estimate the AEs in MV patients. However, these models used an additional clinical protocol and/or invasive measurement, in which they are less clinically feasible [5-7]”. Authors must review in detail the available literature on asynchrony detection in patients under MV. Important references are not quoted an commented (Beitler, Blanch, Thille, de Witt, Chanques,………………..)
Intro. Thrird paragraph. Asynchrony index was defined by Vitacca in Chest 2005 and then further refined by M de Wit (J Crit Care) and Thille, Brochard..(ICM 2006). Colombo used that definitions and Epstein S just commented on it. Please revise.
Intro, page 2, lines 48-67. Since different asynchonies have not been described and lumped together on the unique title of “asynchrony events” it’s very difficult to understand what is the objective of the paper. Asynchronies could be related to the phase of the respiratory cycle or to the intensity of respiratory effort. Both are correct and clinically sound but a global identification under the term AE could be mathematically challenging but very difficult to interpret for clinical use. The question that remains is: could different asynchronies be identified with the method based on time varying elastance?.
Page 2, lines 75-76: “Furthermore, asynchronous breathing is also increasing the MV work of breathing, prolong length of MV and increase the MV [14]”. Reference 14 does not study outcomes, only a mathematical model.
Page 3, lines 93-94: “The patients were sedated using a Puritan Bennet PB840 ventilator”. Please revise grammar construction.
Methods: In 2007 Younes and cols published on measurement of asynchronies based on the equation of motion of the respiratory system. Please expand on how different are both methods. See: A method for monitoring and improving patient: ventilator interaction. Younes M, Brochard L, Grasso S, Kun J, Mancebo J, Ranieri M, Richard JC, Younes H. Intensive Care Med. 2007 Aug;33(8):1337-46. doi: 10.1007/s00134-007-0681-4. Epub 2007 May 31.PMID: 17541554
Results. Please further define the differences between SIMV and spontaneous modes. SIMV could be PCV or VCV.. and all breaths could be triggered by the patients. Then, are spontaneous modes for authors PSV, PAV…..?
Variability is inherent to the human being even if a patient is critically ill under mechanical ventilation. Therefore, some asynchronies during MV are related to the process to breath towards liberation of the MV. Then, all patients will present asynchronies that not per se are associated with a poor outcome. I believe authors can only describe a method to assess in general an asynchronous behavior but not whether or not this interaction of the patient with the MV can cause harm. Moreover, if the message is “There are asynchronies” some less experts attending at bedside will consider increase sedation….. which is a recognized factor that increases mortality. Please comment on all the above.
Discussion, first paragraph: “In this research, the estimation of AEs is further investigated and relates with the type of ventilation modes and also the performance of the lung elastance”. I do not agree with that contention. AE relates with the disease, the ventilator set up by the caregivers and maybe to the ventilator mode. Lung elastance relates with the degree of lung inflammation and asynchronies will be related of the cross-talk between lung receptors and brain (respiratory center). Authors must revise important physiologic concepts. Finally, asynchronies are affected by sedation regimens and this info have not mentioned anywhere.
Page 11, lines 260-261: “Hence, this finding provides evidence that lower values of AEs may indicate a higher elastance value, in which it can further damage lung injury and potentially lead to mortality [25,26]”. This affirmation has no clinical sense. Low AEs could be or not associated with high elastance.
Reviewer 2 Report
The manuscript presents a study that aims to detect and quantify the event of patient-ventilator asynchrony, and authors mentioned the development of software system based on Loo et al. or Poole et al.
I have two major questions,
- The meaning of patient-ventilator asynchrony should be mismatch between patient’s breathing and ventilator support, such as non-trigger, early cycle off, late inspiratory termination etc. In this study, I found this question from figure 3 and 4. Because, in figure 4 exhibited each breathing during mechanical ventilation should be synchrony, although it’s irritable breathing, but inspiratory flow matched to airway pressure. Figure 4 showed that should be only a dyspnea and inconsistence of breathing. However, each breathing is synchrony between patient trigger and ventilator providing. So, it need clarify and revised the manuscript.
- Why the design of study is similar to the Poole’s study? What is the different to Poole et al. of reference15? or specially points of the study and the software?
Minor question
- Whether is the different parameters of ventilator setting in each patient. Why do need collect data in day 2? Have any treatment for patients?
- What is included criteria of patient in this study? Why do select SIMV, PS mode in this study?
Reviewer 3 Report
The paper proposes a non-invasive methodology for noninvasive detection and evaluation of asynchrony events in mechanically ventilated patients. The authors clearly formulate the problem, methods and provide adequate discussion and conclusions. The methods can be used in a regular medical centre.
The design of the research yet can be improved. The patients involved in this study have various clinical diagnoses. I suggest adding a discussion on how it can affect the results (or maybe not? why?).
It may be useful to give a wider background of previous research in modelling lung ventilation. E.g. 10.20428/JST.21.1.1, doi.org/10.3390/computation50100110, 10.1109/I2CACIS49202.2020.9140092 and similar works may be useful in this regard. They may help to develop a more advanced model in the future.
Round 2
Reviewer 1 Report
Authors have appropriately answered my comments
Author Response
Dear respective Reviewer,
We thank the reviewer for his/her constructive comments.This revised manuscript have gone through the professional service of proof read. The certificate is attached here. Please do not hesitate to request additional information, clarification or revision. Thank you for your time and effort in this matter and we look forward to your response.
Regards,
Salwa

Reviewer 2 Report
Despite AE detection being published in the bioengineer field. I still think that the patient-ventilator asynchrony event should have the real detection of patient breathing triggers in the respiratory therapy field, such as esophageal pressure, Edi signal. The present study showed that could say, patient breathing has inconsistent, provide an alert for clinical staff needs further check patient-ventilator asynchrony.
So, I recommend revision in two-part,
- in line 72, Yet, the detection of AEs in MV patients still remain unclear. recommendation: the tool of detection AEs in MV patients is still less.
- In the paragraph of study limitation, should mention that have not patient breathing trigger single to further prove the accuracy of patient-ventilator asynchrony detection.
Author Response
Dear Respective Reviewer,
Please see the attachment.
Many thanks and regards,
Salwa
